# Regulatory Patterns of Crp on Monensin Biosynthesis in *Streptomyces cinnamonensis*

**DOI:** 10.3390/microorganisms8020271

**Published:** 2020-02-17

**Authors:** Chun-Yan Lin, Yue Zhang, Ji-Hua Wu, Rong-Hui Xie, Jianjun Qiao, Guang-Rong Zhao

**Affiliations:** 1Frontier Science Center for Synthetic Biology and Key Laboratory of Systems Bioengineering (Ministry of Education), School of Chemical Engineering and Technology, Tianjin University, Tianjin 300350, China; lcy87828@126.com (C.-Y.L.); zhangy@tib.cas.cn (Y.Z.); 18846321904@163.com (J.-H.W.); rhxie0811@126.com (R.-H.X.); jianjunq@tju.edu.cn (J.Q.); 2Present address: Tianjin Institute of Industrial Biotechnology, Chinese Academy of Sciences, Tianjin 300308, China; 3SynBio Research Platform, Collaborative, Innovation Center of Chemical Science and Engineering (Tianjin), Tianjin 300072, China

**Keywords:** *Streptomyces cinnamonensis*, monensin biosynthesis, Crp, transcription factor, transcription regulation, RNA sequencing

## Abstract

Monensin, produced by *Streptomyces cinnamonensis*, is a polyether ionophore antibiotic widely used as a coccidiostat and a growth-promoting agent in agricultural industry. In this study, cyclic AMP receptor protein (Crp), the global transcription factor for regulation of monensin biosynthesis, was deciphered. The overexpression and antisense RNA silencing of *crp* revealed that Crp plays a positive role in monensin biosynthesis. RNA sequencing analysis indicated that Crp exhibited extensive regulatory effects on genes involved in both primary metabolic pathways and the monensin biosynthetic gene cluster (*mon*). The primary metabolic genes, including *acs*, *pckA*, *accB*, *acdH*, *atoB*, *mutB*, *epi* and *ccr*, which are pivotal in the biosynthesis of monensin precursors malonyl-CoA, methylmalonyl-CoA and ethylmalonyl-CoA, are transcriptionally upregulated by Crp. Furthermore, Crp upregulates the expression of most *mon* genes, including all PKS genes (*monAI* to *monAVIII*), tailoring genes (*monBI*-*monBII-monCI*, *monD* and *monAX*) and a pathway-specific regulatory gene (*monRI*). Enhanced precursor supply and the upregulated expression of *mon* cluser by Crp would allow the higher production of monensin in *S. cinnamonensis.* This study gives a more comprehensive understanding of the global regulator Crp and extends the knowledge of Crp regulatory mechanism in *Streptomyces*.

## 1. Introduction

*Streptomyces* bacteria are a particularly abundant source of natural products, providing more than half of medically valuable antimicrobial and anticancer drugs [1]. The antibiotic biosynthesis in *Streptomyces* is subject to strict control at the transcriptional level by pathway-specific regulators and global regulators [2]. The pathway-specific factors regulate the expression of antibiotic biosynthetic gene cluster and the regulatory patterns on antibiotic biosynthesis are comprehensively established [3]. However, the global regulators exhibit broader effects on both primary and secondary metabolism [4], which might directly or indirectly contribute to the antibiotic biosynthesis in streptomycetes.

The cyclic AMP receptor protein (Crp) is a well-known global factor for regulation of sugar metabolism in *Escherichia coli* [5,6,7]. The nutrient-related effects of Crp on morphology and cell metabolism have been reported in actinomycetes [8]. The *crp* deletion led to the acceleration of sporulation with reduced and delayed germination and other morphological developmental defects in *S. coelicolor* [9,10,11]. In addition to the regulation involved in primary metabolism, previous studies showed that the overexpression of the *crp* gene improved the biosynthesis of actinorhodin, undecylprodigiosin and calcium-dependent antibiotic in *S. coelicolor* and erythromycin in *Saccharopolyspora erythraea* [12,13]. Thus, CRP also plays important regulatory roles in antibiotic biosynthesis.

Monensin, a polyether ionophore antibiotic produced by *Streptomyces cinnamonensis*, is used as a growth-promoting agent for cattle and a coccidiostat for chickens in agricultural industry [14]. In recent years, monensin has been identified as a potential anticancer candidate drug because of its in vitro inhibitory effects on proliferation of several types of cancer cells [15,16,17]. Monensin biosynthetic gene cluster (*mon*) has been sequenced, and the functions have been partially characterized [18,19]. Eight separate type I polyketide synthases (PKSs) (MonAI to MonAVIII) were proposed to be responsible for assembling short chain acyl-CoA precursors into carbon backbone of monensin molecule [20]. The tailoring enzymes MonBI, MonBII and MonCI are involved in the oxidative cyclization of a linear polyketide intermediate [21,22,23]; MonD hydroxylases at C-26, and MonE methylates hydroxyl group at C-3 [24]. Thioesterase releases the final monensin from polyketide synthase [25].

Our laboratory focuses on the transcriptional factors for regulation of monensin biosynthesis. Our previous study showed that MonRI, MonRII and MonH are three positive pathway-specific regulators and that they cooperatively control *mon* genes in monensin biosynthesis [26]. The global factor DasR positively controls monensin biosynthesis via regulating both pathway-specific regulatory gene *monRII* and function genes such as *monAIX*, *monE* and *monT* [27]. In this study, we endeavored to decipher Crp regulatory patterns in monensin biosynthesis and identified several key genes in acyl-CoA supply pathways and most of the *mon* genes that are subjected to Crp regulation.

## 2. Materials and Methods

### 2.1. Bacterial Strains, Chemicals and Growth Conditions

The parental strain *S. cinnamonensis* ST021, which is derived from *S. cinnamonensis* ATCC15413, was deposited in China General Microbiological Culture Collection Center (CGMCC, Beijing, China) under the accession number CGMCC 7.240. Modified Gauze’s Medium No.1 (20 g/L starch, 5 g/L soybean flour, 1 g/L KNO_3_, 0.5 g/L NaCl, 0.5 g/L MgSO_4_, 0.5 g/L K_2_HPO_4_, 0.01 g/L FeSO_4_, 18 g/L agar, pH adjusted to 7.4) was employed for the growth and sporulation of *S. cinnamonensis* strains for 7 days under 30 °C. For fermentation, 25 mL of seed medium (16 g/L soybean flour, 2.5 g/L yeast extract, 20 g/L dextrin, 5 g/L glucose, 1.2 g/L CaCO_3_, pH adjusted to 6.8) was contained in a 250-mL flask and cultivated for 24 h under 30 °C and 250 rpm. Fermentation medium (25 g/L soybean oil, 36.7 g/L glucose, 16 g/L soybean flour, 3 g/L CaCO_3_, 2 g/L Na_2_SO_4_, 2.2 g/L NaNO_3_, 0.1 g/L FeSO_4_, 0.07 g/L Al_2_ (SO_4_)_3_, 0.075 g/L K_2_HPO_4_, 0.01 g/L vitamin C, pH adjusted to 6.8) was introduced with 8% volume of seed culture and cultivated at 30 °C and 250 rpm for 10–12 days. All media were sterilized by autoclaving at 121 °C for 20 min before use. When necessary, antibiotics were added in the following concentrations: 25 μg/mL hygromycin, 20 μg/mL nalidixic acid.

### 2.2. Plasmid Constructions for Gene Overexpression and Silencing

The plasmids and strains used in this study are listed in Table 1. The primers are listed in Appendix A. Gene manipulation of *Escherichia coli* and *Streptomyces* were performed according to standard protocols [28].

DNA fragment of the *crp* coding section was amplified from *Streptomyces cinnamonensis* genome DNA, using CZ-F and CZ-R as primers, and integrated into pLCY009 between *Nde* I and *Xba* I sites, generating the integrative *crp* overexpression vector pLCY009-*crp.* The nucleotide sequence of the *crp* gene was deposited under the GenBank accession number KY305484.

Antisense RNA vector was constructed to hinder the translation of the *crp* gene. A 120-bp reverse complementary sequence to 60-bp 5′ untranslation region (UTR), RBS and 60-bp coding sequence was designed as the binding sequence of the *crp* mRNA. This binding sequence was flanked with one restrict site and 21-bp inverted repeat overhangs to form a hairpin in the 5′ and 3′ ends [29]. This 174-bp DNA fragment containing binding part and hairpin part was amplified using primers CS1-F and CS1-R, and then was integrated into pLCY010 between *Hind* III and *Xba* I sites, generating the replicative anti-*crp* RNA vector pLCY010-csRNA.

### 2.3. Construction of Engineered S. cinnamonensis Strains

The recombinant plasmids were transferred into *S. cinnamonensis* via conjugation with the help of *E. coli* ET12567. Overexpression strain ST042 and anti-*crp* strain ST043 were generated from strain ST021 introduced with pLCY009-*crp* and pLCY010-csRNA, respectively.

### 2.4. RNA-Seq Library Preparation, Clustering and Sequencing

RNA was extracted from 6-day mycelia in fermentation medium, and 3 μg RNA was used to prepare each sample. NEBNext Ultra Directional RNA Library Prep Kit for Illumina (NEB, Ipswich, MA, USA) was used to generate sequencing libraries according to the manufacturer’s protocol. The library quality was assessed with the Agilent Bioanalyzer 2100 system.

The clustering of the index-coded samples was performed on a cBot Cluster Generation System using TruSeq PE Cluster Kit v3-cBot-HS (Illumina, San Diego, CA, USA) following the manufacturer’s recommendation. Illumina Hiseq platform was employed to sequence the library and generate paired-end reads. The transcriptome data of ST021, ST042 and ST043 have been deposited in the NCBI Sequence Read Archive (SPA) under accession number PRJNA398614.

### 2.5. Quantification and Differential Analysis of Gene Expression

The reads number mapped to each gene was counted by HTSeq v0.6.1. FPKM (expected number of Fragments Per Kilobase of transcript sequence per Million base pairs sequenced) is the most commonly used method for estimating gene expression levels [31], and it was calculated based on the length of each gene and read counts mapped to one gene.

The read counts of each library were adjusted by edgeR program package through one scaling normalized factor. DEGSeq R package was employed to analyze differential expression of different conditions [32]. The *p*-values were adjusted according to the Benjamini and Hochberg method. The *q*-value (adjusted *p*-value) of 0.005 and log2 (Fold change) of 1 were set as the threshold for significantly differential expression.

### 2.6. GO and KEGG Enrichment Analysis of Differentially Expressed Genes

GOseq R package was used to analyze gene ontology (GO) enrichment of differentially expressed genes (DEGs). Gene length bias was corrected in the final results. GO terms with *q*-values under 0.05 were considered to be significantly enriched. Statistical enrichment of DEGs in KEGG pathways was tested by KOBAS software. Model *Streptomyces coelicolor* was chosen as the reference microorganism.

### 2.7. Semi-Quantitative RT-PCR

Primers involved in RT-PCR experiments are listed in Appendix A. RNA was isolated from the 6-day *S. cinnamonensis* mycelia in fermentation medium. Reverse transcription was conducted employing First-Strand cDNA Synthesis SuperMix (TransGen Biotech, Beijing, China) using 500 ng of total RNA as template. The resultant cDNAs, diluted five times, were used as template in PCR analysis. PCR conditions were set as follows: 95 °C for 5 min; 30 cycles of 95 °C for 5 s, 55–60 °C for 30 s and 72 °C for 30 s; 72 °C for 10 min. PCR products were separated by agarose gel electrophoresis and exposed under UV to calculate their relative intensities by densitometric analysis software. All data were verified by three independent experiments.

### 2.8. Extraction and Analysis of Monensin

Monensin was extracted from mycelium cells and the titer was measured by HPLC [27]. In detail, mycelium cells of *S. cinnamonensis* were separated from fermentation broth after centrifugation, washed with PBS three times and resuspended in methanol. After ultrasonic cytolysis for 1 h, the lysate was centrifuged and the supernatant was filtered with 0.22 μm membrane and used to evaluate monensin titer. Analysis was carried out by HPLC using an Agela 5 μm ODS3 100 Å column (250 mm × 4.6 mm), and the titer was quantified by Alltech ELSD 2000 evaporative light scattering detector. The mobile phase was composed of solvent A (20 mM ammonium acetate) and solvent B (methanol), and the linear gradient program was set as follows: 80%–100% solvent B (0–25 min) and 100%–80% solvent B (25–30 min). The flow rate was at a constant 1 mL/min.

## 3. Results

### 3.1. Crp Positively Regulates Monensin Biosynthesis

To evaluate the regulatory role of Crp in monensin biosynthesis, the *crp* overexpression strain ST042 and the anti-*crp* stain ST043 with anti-*crp* sequences (Figure 1a) were constructed, and monensin yield and mycelium dry weight (MDW) at intervals of two days throughout fermentation were detected. The mycelium growth was early stimulated by overexpressing the *crp* gene and retarded by silencing the *crp* gene (Figure 1b), implying that Crp has a positive effect on the vegetative development of *S. cinnamonensis*. Furthermore, compared to the original strain ST021 (14.9 mg monensin/mg MDW) (Figure 1c), strain ST042 produced 19.6 mg monensin per mg MDW after 10-day fermentation with 31.5% increase, while strain ST043 produced 8.18 mg monensin per mg MDW with 45.1% reduction. The results indicated that Crp positively regulates monensin biosynthesis, which was not due to the mycelium growth.

### 3.2. RNA-Seq Shows Global Regulatory Roles of Crp

To gain insight into the regulatory mechanism of Crp on monensin biosynthesis, RNA sequencing was conducted. Totals of 15.7, 20.6 and 22.7 million clean reads from strain libraries of ST021, ST042 and ST043 were obtained after filtration of 16.5, 21.2 and 23.3 million raw reads, respectively, responding for more than 93.94% at the Q20 level (sequence error probability of 0.04%). These clean reads were mapped to the *S. cinnamonensis* ST021 genome, resulting in 15.6 (99.29%), 17.9 (86.75%) and 20.7 (91.16%) million mapped reads for ST021, ST042 and ST043, respectively (Table 2).

The levels of gene expression were evaluated by FPKM values. On the basis of the applied criteria (*q-value* < 0.005 and fold changes ≥ 2), DEGs were identified among strains ST021, ST042 and ST043. When compared to the original strain ST021, 756 genes were upregulated and 645 genes were downregulated in the *crp* overexpression strain ST042 (Figure 2a), while 1237 genes were upregulated and 1326 genes were downregulated in the anti-*crp* strain ST043 (Figure 2b). Additionally, strains ST042 and ST043 shared 820 DGEs compared to ST021 (Figure 2c). Particularly, the expression levels of the *crp* gene in three strains were evaluated. The FPKM values of the *crp* gene were 418.32 and 116.51, in the *crp* overexpression strain ST042 and the anti-*crp* strain ST043, representing 2.2-fold enhancement and 38.3% decrease, respectively, compared to the original strain ST021, in which FPKM value of the *crp* gene was 188.91. These results suggested the extensive effects of Crp on the regulation of gene expression in *S. cinnamonensis*. A total of 3144 genes were differentially expressed under the conditions of the overexpression and partial silencing of the *crp* gene.

To investigate the functions of DEGs influenced by Crp, gene ontology (GO) enrichment was used for describing the biological roles of genes and their coding proteins [33]. All genes were classified into function groups in three main categories (Figure 3). In the biological process category, gene expression (GO: 0010467), cellular protein metabolic process (GO: 0044267) and translation (GO: 0006412) were significantly enriched. Nine GO terms including intracellular part (GO: 0044424), cytoplasm (GO: 0005737) and organelle (GO: 0043226) were significantly enriched in the cellular component category. In the molecular function category, structural molecule activity (GO: 0005198), RNA binding (GO: 0003723) and structural constituent of ribosome (GO: 0003735) were significantly enriched. These results further revealed the broad regulatory effects of Crp on various intracellular functions in *S. cinnamonensis*.

### 3.3. Crp Upregulates Biosynthetic Pathways of Monensin Precursors

Malonyl-CoA, methylmalonyl-CoA and ethylmalonyl-CoA are precursors for monensin biosynthesis and are derived from primary metabolism. To decipher the genes involved in the precursor biosynthesis, the 777 DEGs were mapped to the 92 KEGG pathways in the database. The majority of the 57 DEGs involved in glycolysis, fatty acid degradation and other carbon metabolisms (Table 3, Appendix A) were transcriptionally promoted by the *crp* overexpression in strain ST042 and inhibited by the *crp* partial silencing in strain ST043 (Figure 4a). The 10 DEGs enriched in glycolysis were responsible for acetyl-CoA formation from glucose via pyruvate metabolism and consequently in increasing the biosynthesis of malonyl-CoA. In fatty acid degradation, most of the 20 DEGs code acyl-CoA dehydrogenases, acetyl-CoA acetyltransferases and fatty-acid-CoA ligases. In carbon metabolism, the 27 DEGs are involved in the biosynthesis of butyryl-CoA and ethylmalonyl-CoA from crotonyl-CoA and of methylmalonyl-CoA from the intermediate succiny-CoA of the TCA cycle, which would enhance the supply of acyl-CoA building blocks in monensin biosynthesis.

In order to identify the DEGs responding for biosynthesis of precursors malonyl-CoA, methylmalonyl-CoA and ethylmalonyl-CoA, several representative genes in those metabolic pathways were chosen and reverse-transcription PCR was performed with specific primers listed in Appendix A. Compared to the original strain ST021, the transcriptional levels of *acs* (Scin0389), *pckA* (Scin4931), *acdH* (Scin4869), *atoB* (Scin7189) and *accB* (Scin4012) genes in malonyl-CoA biosynthetic pathways and *mutB* (Scin5352), *epi* (Scin6197) and *ccr* (Scin6441) genes in methylmalonyl-/ethylmalonyl-CoA biosynthetic pathways were increased 1.63-fold to 3.34-fold in strain ST042 (Figure 4b) and decreased by 32% to 65% in strain ST043. The results of RT-PCR analysis were comparable to RNA-Seq data (Figure 4b), confirming that Crp enhanced monensin biosynthesis via upregulating biosynthesis of the short chain acyl-CoA precursors. In addition, time-course transcription analysis of the *crp* gene in strains ST021, ST042 and ST043 was conducted. At different intervals in the fermentation course, the transcriptional levels of the *crp* gene were constantly higher in ST042 and lower in ST043 than those in ST021, which was consistent with the results of RNA sequencing (Figure 4c).

### 3.4. Crp Upregulates Transcription of mon Cluster

There are twenty genes in the *mon* cluster, and our previous study suggested that the monensin biosynthetic gene cluster is transcribed into eight transcription units (TUs) [26], shown in Figure 5a. The RNA-seq analysis results showed that TU4, TU5, TU6, TU7 and TU8 in the *mon* cluster were significantly upregulated in the *crp* overexpression strain ST042 and correspondingly downregulated in the anti-*crp* strain ST043, compared to the original strain ST021 (Figure 5b). In contrast, TU2 was downregulated in ST042 and ST043. The transcription of TU1 was decreased in ST043 and not obviously changed in ST042, while the transcription of TU3 was increased in ST042 and not obviously changed in ST043. Thus, RT-PCR was further conducted and the transcripts of *mon* TUs were assessed. When *crp* was overexpressed, the transcripts of TU4, TU5, TU6, TU7 and TU8 were increased by 2.1-fold to 4.8-fold compared to the original strain ST021. In contrast, partial *crp* silencing resulted in 34.0% to 71% decrease of transcripts of these five TUs (Figure 5c). No significant changes in the transcripts of TU1, TU2 and TU3 were detected by RT-PCR whenever the *crp* gene was overexpressed or partially inhibited (Figure 5c). Taken together, Crp positively upregulated the transcription of five *mon* TUs, including PKS genes, tailoring genes and pathway-specific regulatory gene *monRI*.

## 4. Discussion

In this work, the forward and reverse genetic experiments were conducted, the differential gene expressions were analyzed by RNA sequencing and RT-PCR and the global positive regulation of Crp on monensin biosynthesis in *S. cinnamonensis* was deciphered. Crp enhanced the expression of the crucial genes involved in the biosynthesis of acetyl-CoA, malonyl-CoA, methylmalonyl-CoA and ethylmalonyl-CoA. In the *crp* overexpression strain ST042, the DEGs enriched in glycolysis were mainly responsible for glucose utilization and pyruvate metabolism, which would consequently increase the supply of acetyl-CoA. In fatty acid degradation, the majority of the DEGs coded acyl-CoA dehydrogenases, which were involved in each step of long-chain fatty acid degradation. Furthermore, genes that coded acetyl-CoA acetyltransferases and long chain fatty acid CoA ligases were also transcriptionally promoted by the *crp* overexpression. In carbon metabolism, the products of the DEGs were involved in a variety of reactions, including the synthesis of malonyl-CoA from acetyl-CoA, the metabolism of the intermediate succiny-CoA in TCA cycle to methylmalonyl-CoA and the conversion of crotonyl-CoA to butyryl-CoA and ethylmalonyl-CoA, which would enhance the supply of acyl-CoA building blocks in monensin biosynthesis. Furthermore, the majority of genes in the *mon* cluster were also upregulated by Crp.

The sufficient availability of precursors, which are supplied from primary metabolism, is a prerequisite for the biosynthesis and production of secondary metabolites [34]. In monensin biosynthesis, five malonyl-CoA, seven methylmalonyl-CoA and one ethylmalonyl-CoA were proposed to be loaded by thirteen acetyltransferase (AT) domains of *mon* PKSs and used as extender units in the carbon backbone formation of one monensin A molecule [19,35,36]. Acetyl-CoA is the central carbon node connecting precursor supply from primary pathways with monensin biosynthesis. In oil-rich fermentation medium, the fatty acid degradation pathway can provide acetyl-CoA for monensin biosynthesis [37]. Additionally, when glucose is utilized for mycelium growth, acetate is the major byproduct, and acetyl-CoA synthetase (ACS) is responsible for the synthesis of acetyl-CoA from acetate. The corresponding genes of *acs* in the glycolytic pathway and *acdH* (coding acyl-CoA dehydrogenase) and *atoB* (coding acetyl-CoA acetyltransferase) in the fatty acid degradation pathway were upregulated by Crp, indicating that the formation of acetyl-CoA is enhanced to improve monensin production in *S. cinnamonensis*.

The canonical extender unit of polyketide biosynthesis, malonyl-CoA, is mainly derived from carboxylation of acetyl-CoA by acetyl-CoA carboxylases (ACC) [38,39]. Overexpression of *acc* genes led to significant increase in the production of polyketides actinorhodin in *S. coelicolor* [40] and mithramycin in *S. argillaceus* [41]. When the *crp* gene was overexpressed in *S. cinnamonensis*, the transcription of the *accB* gene was upregulated, which would increase malonyl-CoA supply for monensin biosynthesis.

As seven methylmalonyl-CoA extender units are required to build a single polyketide of monensin, the supply of methylmalonyl-CoA may represent a limiting step in the biosynthesis of monensin [36]. Succinyl-CoA stemming from the TCA cycle was sequentially catalyzed by methylmalonyl-CoA mutase (Mut) and methylmalonyl-CoA epimerase (Epi) to (*2R*)- then to (*2S*)-methylmalonyl-CoA [42]. Phosphoenolpyruvate is converted into oxaloacetate by phosphoenolpyruvate carboxykinase (PckA), which allows carbon flux from glucose toward the TCA cycle to strengthen the formation of succinyl-CoA. Additionally, crotonyl-CoA reductase (CCR) was proved to be essential for providing the majority of methylmalonyl-CoA for monensin biosynthesis and when *S. cinnamonensis* was grown on an oil-based medium [37,43]. Acetyl-CoA is orderly transformed into acetoacetyl-CoA by AtoB then into crotonyl-CoA, which will be subsequently catalyzed to butyryl-CoA by CCR. Butyryl-CoA pool can be used to generate not only ethylmalonyl-CoA but also methylmalonyl-CoA precursor pool under oil-based nutrient condition [37]. The transcription of *pckA*, *mutB*, *epi* and *ccr* were upregulated when the crp gene was overexpressed in strain ST042, which would result in more efficient biosynthesis of methylmalonyl-CoA and higher monensin titer.

Most importantly, Crp upregulates five transcription units in the *mon* cluster, which cover PKSs genes, tailoring genes and pathway-specific regulatory gene. In *S. coelicolor*, Crp regulated 8 out of 22 predicted secondary metabolic clusters, and its regulatory targets were composed of PKSs and tailoring genes in *act*, *red*, *cda* and *ycpk* clusters [12]. The *monAI*-*monAVIII* genes coding the PKSs are responsible for assembling precursors into the monensin backbone [44]. Then, tailoring enzyme MonCI mediates an epoxidation of the polyolefin constructed by polyketide synthases, and MonBI and MonBII cooperatively catalyze the cyclization of the resultant polyepoxide to yield the polyether skeleton of monensin. Our previous studies showed that three pathway-specific regulators (MonRI, MonRII and MonH) [26] and the global regulator DasR [27] do not regulate the transcriptions of *mon* PKSs genes and tailoring genes *monCI*, *monBI* and *monBII*. Here, our results suggested that the targets of Crp regulation would be the PKSs and *monCI*, *monBI* and *monBII* genes of the *mon* cluster. Moreover, Crp regulations on pathway-specific regulatory genes *actII-ORF4*, *redZ* and *cdaR* were identified in *S. coelicolor* [12]. Of three pathway-specific regulators for monensin biosynthesis, MonRI and MonRII upregulated the transcription of tailoring gene *monD* and thioesterase gene *monAX* [26], while *monRI* was under the control of Crp, indicating that Crp possibly exerted regulatory effects on the transcription of *monD* and *monAX* through its tuning on *monRI* transcription.

## 5. Conclusions

A global regulatory model of Crp on monensin biosynthesis in *S. cinnamonensis* was proposed in this study (Figure 6). Crp upregulated curial primary carbon metabolic genes involved in glycolysis (*pckA, acs*) and fatty acid degradation (*atoB*, *acdH*), together with the genes involved in biosynthesis of malonyl-CoA (*accB*), methylmalonyl-CoA (*mutB*, *epi* and *ccr*) and ethylmalonyl-CoA (*ccr*), which might make great contribution to precursor supply in monensin biosynthesis. Crp also upregulated the transcription of *mon* cluster in multiple patterns, including enhancing the expression of PKSs genes, tailoring genes (*monBI*-*monBII-monCI, monD* and *monAX*) and a pathway-specific activator gene *monRI*. Our work sheds new light on the regulatory mechanism of Crp on monensin biosynthesis and extends the understanding of Crp functions in *Streptomyces*.

## Figures and Tables

**Figure 1 microorganisms-08-00271-f001:**
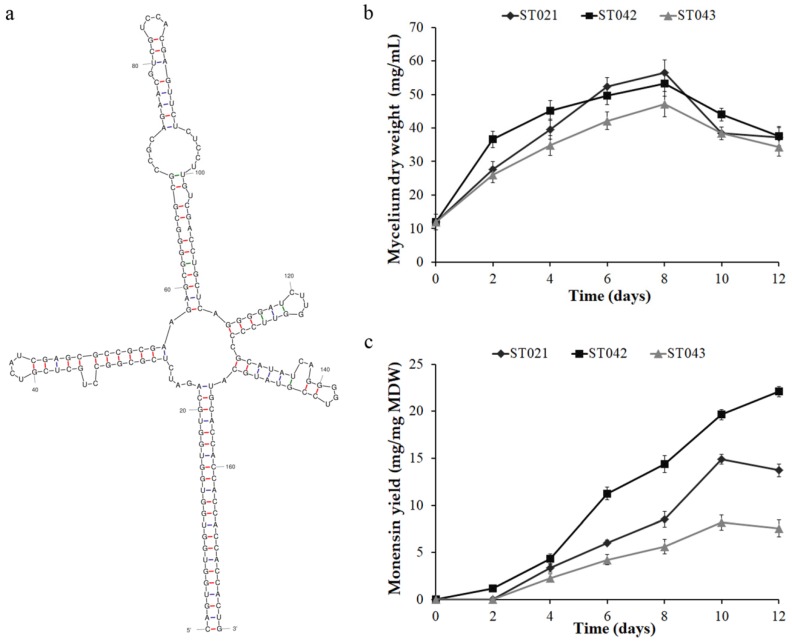
The overexpression and antisense RNA silencing of the *crp* gene. (**a**) The predicted secondary structure of anti-*crp* sequences on pLCY010-csRNA vector. Mycelium dry weight (MDW) (**b**) and monensin yield (**c**) of the original strain ST021, the crp overexpression strain ST042 and the anti-*crp* strain ST043 of *S. cinnamonensis*. Error bars indicate standard deviations from three independent experiments.

**Figure 2 microorganisms-08-00271-f002:**
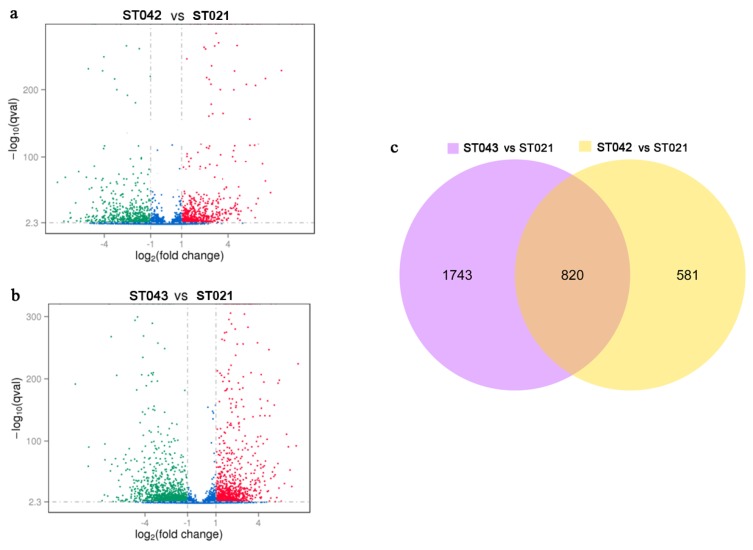
The volcano plots and Venn diagram of differential expressed genes (DEGs) in strains ST021, ST042 and ST043. Volcano plots of ST042 vs. ST021 (**a**) and ST043 vs. ST021 (**b**). DEGs are shown in red (upregulated) and green (downregulated) points, while the others are shown in blue points. The horizontal axis represents the fold change of gene expression in different samples, while the vertical axis represents the statistical significance of gene expression difference. (**c**) Venn diagram of DEGs. Compared with ST021, the number of DEGs in ST043 is shown in the purple circle on the left, while that in ST042 is in orange circle on the right. The intersection of the two circles represents the number of DEGs shared by ST042 and ST043.

**Figure 3 microorganisms-08-00271-f003:**
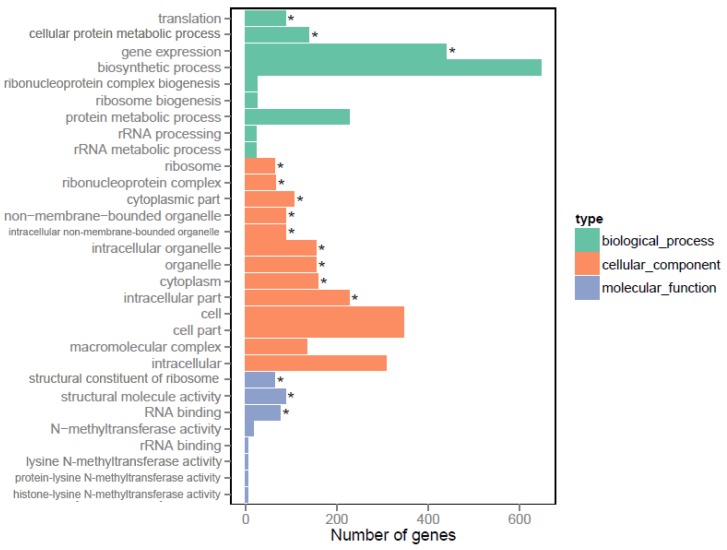
Functional categorization of DEGs in gene ontology (GO) in strain ST042. The asterisks indicate significantly enriched GO terms.

**Figure 4 microorganisms-08-00271-f004:**
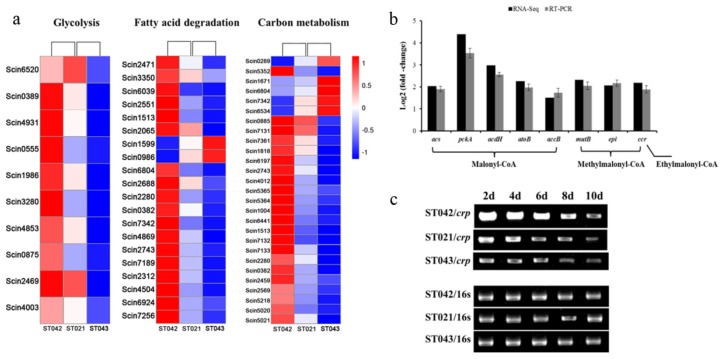
Transcription analysis of selected primary metabolic genes. (**a**) Heat maps of 57 DEGs enriched in the processes of glycolysis (10), fatty acid degradation (20) and carbon metabolism (27). Gene names involved are listed beside each heat map. (**b**) The relative transcriptional levels of primary metabolic genes in precursor biosynthesis in the *crp* overexpression strain ST042 by RNA-Seq and RT-PCR. Those in the original strain ST021 were set to 1.0 (arbitrary units). *acs*, acetyl-CoA synthetase gene; *pckA*, phosphoenolpyruvate carboxykinase gene; *acdH*, acyl-CoA dehydrogenase gene; *atoB*, acetyl-CoA acetyltransferase gene; *accB*, acyl-CoA carboxylase complex B subunit gene; *mutB*, coenzyme B12-dependent methylmalonyl-CoA mutase B subunit gene; *epi*, methylmalonyl-CoA epimerase gene; *ccr*, crotonyl-CoA reductase gene. Error bars indicate standard deviations from three independent experiments. (**c**) The *crp* transcription of ST042, ST021 and ST043 at different intervals.

**Figure 5 microorganisms-08-00271-f005:**
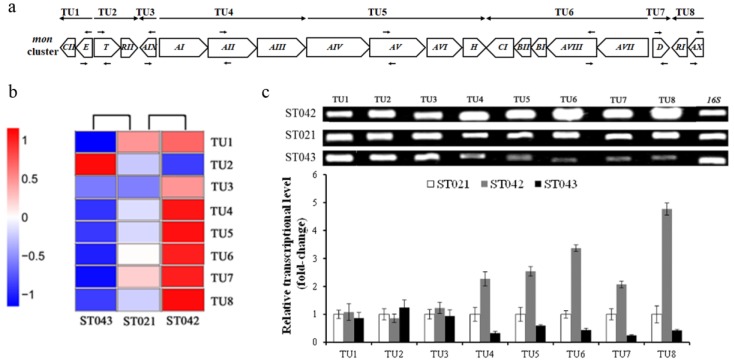
RNA-Seq analysis and RT-PCR verification for transcription of *mon* cluster. (**a**) Transcription units (TUs) in monensin biosynthesis. (**b**) Heat map of clustering analysis of eight *mon* TUs by RNA-seq. Expression ratios are shown as log2 values. Red color represents increased expression, while blue color represents decreased expression. (**c**) Relative transcriptional levels of TUs in strains ST042 (grey column) and ST043 (black column) compared to strain ST021 (white column), which were set to 1.0 (arbitrary units) by RT-PCR. Error bars indicate standard deviations from three independent experiments.

**Figure 6 microorganisms-08-00271-f006:**
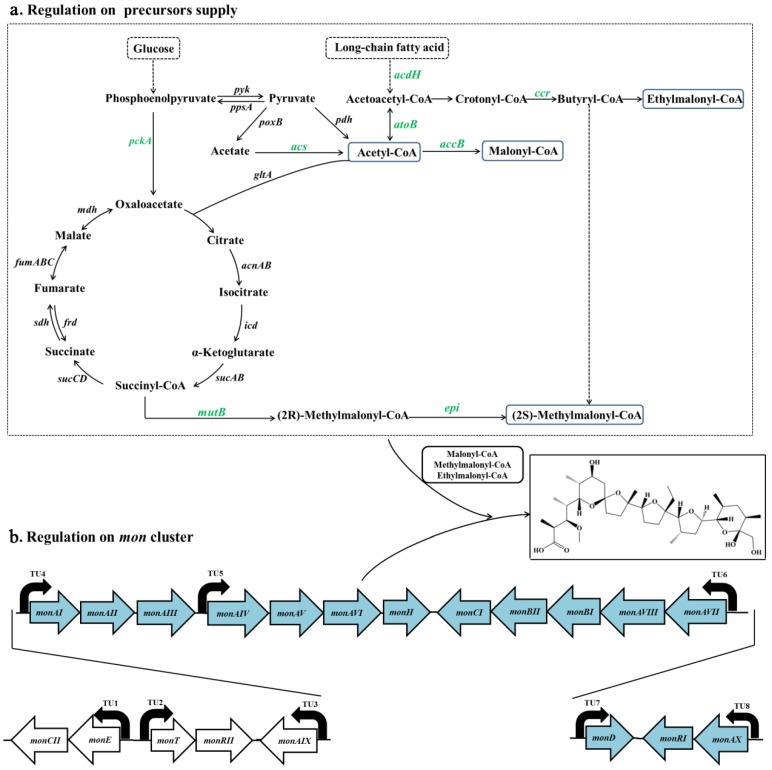
Proposed regulatory mechanism of Crp for monensin biosynthesis. (**a**) The primary metabolism of precursor supply for monensin biosynthesis. The metabolic genes under Crp regulation are shown in green. (**b**) The monensin biosynthetic gene cluster. The *mon* genes of the transcription units (TUs) under Crp regulation are shown in blue. Curved black arrows represent transcription directions. *pyk*, pyruvate kinase gene; *ppsA*, phosphoenolpyruvate synthase gene; *poxB*, pyruvate oxidase gene; *pdh*, pyruvate dehydrogenase gene; *acnAB*, aconitate hydratase gene; *icd*, isocitrate dehydrogenase gene; *sucABCD*, succinyl CoA synthase gene; *frd*, fumarate reductase gene; *sdh*, succinate dehydrogenase gene; *fumABC*, fumarase ABC gene; *mdh*, malate dehydrogenase gene; *gltA*, citrate synthase gene.

**Table 1 microorganisms-08-00271-t001:** Bacterial strains and plasmids used in this study.

Strains/Plasmids	Characteristics	Reference or Source
**Strains**		
*Escherichia coli*		
DH5α	Recipient for cloning experiments	Invitrogen
ET12567/pUZ8002	Strain for conjugal transfer, with pUZ8002 helper plasmids, Chl^R^, Kan^R^	[28]
*Streptomyces cinnamonensis*		
ST021	The parental strain derived from *Streptomyces cinnamonensis* ATCC15413	This work
ST042	ST021 with pLCY009-*crp* integrative expression vector, Hyg^R^	This work
ST043	ST021 with pLCY010-csRNA replicative anti-*crp* RNA vector, Hyg^R^	This work
**Plasmids**		
pLCY009	pIB139 derivative, ΦC31 attachment site, integrative, carrying P_ermE*_ promoter, Hyg^R^	[27]
pLCY010	pUWL201 derivative, pIJ101 replicon, replicative, carrying P_ermE*_ promoter, Amp^R^, Tsr^R^, Hyg^R^	[30]
pLCY009-*crp*	*crp* overexpression vector, ΦC31 attachment site, integrative, carrying P_ermE*_ promoter, Hyg^R^	This work
pLCY010-csRNA	Anti-*crp* RNA vector, pIJ101 replicon, replicative, carrying P_ermE*_ promoter, Hyg^R^	This work

Amp^R^, ampicillin resistance; Chl^R^, chloramphenicol resistance; Hyg^R^, hygromycin resistance; Kan^R^, kanamycin resistance; Tsr^R^, thiostrepton resistance; P_ermE*_, the contitutive ermE promoter.

**Table 2 microorganisms-08-00271-t002:** Statistical analysis of RNA sequence data and genome mapping.

Samples	ST021	ST042	ST043
Raw reads	16,537,074	21,162,804	23,277,776
Clean reads	15,709,834	20,599,352	22,714,616
Clean bases (G)	1.96	2.57	2.84
GC content (%)	65.57	63.56	66.01
Q20 Base ratio (%) *	93.94	97.27	97.26
Q30 Base ratio (%)	88.06	93.11	93.05
Mapped reads	15,597,581	17,869,047	20,707,208
Mapped reads ratio (%)	99.29	86.75	91.16

* Q20 Base ratio: error probability of 1%. Q30 Base ratio: error probability of 0.1%.

**Table 3 microorganisms-08-00271-t003:** Selected KEGG enrichment results of DEGs between strains ST042 and ST021.

Pathway ID	Pathway	Number of DEGs	Representative Genes *
sco00010	Glycolysis	10	*acs*, *pckA*
sco00071	Fatty acid degradation	20	*acdH*, *atoB*
sco01200	Carbon metabolism	27	*accB*, *mutB*, *epi*, *ccr*

* The putative functions of genes involved are listed in Appendix A.

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
