# Peer review of "Regulatory Patterns of Crp on Monensin Biosynthesis in Streptomyces cinnamonensis"

_microorganisms, 2020, doi:10.3390/microorganisms8020271_

Round 1

Reviewer 1 Report

The work is very well explained and figures supporting are very clear. 

As a minor comment or suggestion, figure 1 could be done as a combination of a and b showing that the levels of production are also due to the bacterial growth curve i.e. only one curve in which we have mg monensin/mg dry weight.

As a mayor comment, a RT-PCR is not a quantitative method, so for comparing with RNA-seq data this has to be executed by quantitative real-time PCR (qRT-PCR). In my opinion the densitometric analysis is not reliable enough to be comparable with the RNA-seq reads. So this part of the work should be repeated with real quantitative data.

Reviewer 2 Report

The work is very interesting and in my opinion covers comprehensively and adequately the objects set. The discussion is very good and in combination with the results they produce a final result that will provide valuable help to those who are involved in the subject being dealt with. Therefore I suggest the publication as it is with only minor changes relating to the use of English.

Reviewer 3 Report

In the publication “Regulatory patterns of Crp on monensin biosynthesis in Streptomyces cinnamonensis” the authors experimentally created and fully characterized a new microorganism like Streptomyces cinnamonensis bacteria. It is useful in the effective production of Monensin, a growth promoting agent in agricultural industry.

In my opinion, this work can be published in the Journal Microorganisms after removing a number of shortcomings and hindering her reading, listed below:

1) At line 44 the second paragraph in the first subsection was unfortunately started with an acronymis: … Crp, …

Do you mean: Cyclic reactive protein?

If so, please correct for: … The cyclic AMP receptor protein (Crp), …, or please define “Crp”

2) At line 54 is: … in-vitro …

Should be: … in vitro ..., such as, for example, in publication Microorganisms 2019, 7, 286; doi:10.3390/microorganisms7090286 in page 15 at second line, fourth paragraph, page 16 at the second line and at eighth line from below, and in References 9, 58, 59, 74 and 92 cited therein.

The sign “-“ is unnecessary and the text borrowed from another language in this case, we usually write in italics style.

3) At lines 58, 89, 194, 291, 301, and 302 please enter missing spaces before the reference to the source literature.

4) At line 143 is: … (250×4.6 mm), …, but should be: … (250 × 4.6 mm), … . The mathematical sign of multiplication “× “ requires space characters before and after it, generally.

5) At lines 145 and 146 author have repeatedly entered a small "-" pause between numbers, and author should generally use the average space "–". Is … 80-100% … … 0-25 … … 100-80% … … 25-30 min …, but should be: … 80–100% … … 0–25 … … 100–80% … … 25–30 min …, respectively.

Similarly, please correct the pause sign “-“ for the average pause “–“ sign between page numbers in the literature cited in the lines: 369, 371, 373, 376, 379, 384, 386, 389, 392, 396, 401, 403, 406, 410, 416, 420, 425, 428, 431, 434, 437, 440, 443, 451, 456, 460, 464, 466, 470, 472, 474, 476, 479, etc.

6) In line 158 in Figure 1a and Figure 1b the unit of volume (L) on the ordinate (y) is in capital letters, as in line 146, and in paragraph 2.1. (many times), but on line 152 it is given a different style twice.

Please unify.

7) In line 158 in Figure 1 unit of time “(d)”  is correctly entered in the abscissa (x), but the description of images in lowercase can mislead the reader. Please break the system and enter the word “(days)” instead of “(d)”.

8) At line 194 from the version of the publication to another editor there is a link to the source literature: … (Young at al., 2010) … . Please delete … (Young at al., 2010) … because the citation is doubled.

9) At lines 234 and 262 (at Figure 4 and Figure 5) the drawings are marked with capital letters of the alphabet, but in figures 1 and 2 (lines 158 and 184) the drawings are marked with small letters of the alphabet. Please make it clear with captions under the figures and in the text. Please also harmonize the uppercase or lowercase letters style, such at line 237 is …(B) …, but at line 235 is … (A) …, and at line 244 is … (C) … .

10) At line 309 is: … (2R)- then to (2S)- …, but should be: … (2R)- then to (2S)- … .

“Rectus” and “Sinister” symbols "R" and "S", respectively, should be in italics.

11) Line 366, References.

In magazine names, words start with uppercase or lowercase letters accidentally and they should always start with a capital letter!

Also in the title, the second and subsequent words begin with a lowercase letter, and the authors break this rule, see the lines such: 385 and 388.

12) At line 339 in Conclusion at line 2 is at bold style: … Figure 6 … . 

Please correct for a normal style as in other similar cases as in the lines: 259, 257, 251, 248, 231, 226, 225, 212, 221, 195, 210, etc.

13) Line 427 is: … Tetrahedron letters …, but should be: … Tetrahedron Lett. … .

As a general rule, all references should include the generally accepted short name of the Journal, if any

14) The Supplementary Material in Table S1 contains references to the source such Kieser et al. 2000, Zhang et al. 2016 and Pang et al. 2015, but they are not complete. Please provide references paragraph at the end and provide full references.

Alternatively, this quite important Table S1 could be moved to the main work.

15) In the Supplementary Materials, the authors included an extremely transparent Figure S1.

This figure, after moving to the main work, after rewriting into a slightly compressed form, would seriously facilitate the study of this extensive experimental material by the reader.

Round 2

Reviewer 1 Report

The authors have improved the manuscript as it was requested.